# Development of a Magnetoresistive-Based Wearable Eye-Tracking System for Oculomotor Assessment in Neurological and Otoneurological Research—Preliminary In Vivo Tests

**DOI:** 10.3390/brainsci13101439

**Published:** 2023-10-10

**Authors:** Aniello Donniacuo, Francesca Viberti, Mario Carucci, Valerio Biancalana, Lorenzo Bellizzi, Marco Mandalà

**Affiliations:** 1Department of Medicine, Surgery and Neuroscience, University of Siena, U.O.C. Otorinolaringoiatria, Viale Bracci 16, 53100 Siena, Italy; nellodonniacuo@gmail.com (A.D.); carucci.mario@libero.it (M.C.); mandal@unisi.it (M.M.); 2Department of Physical Sciences, Earth and Environment, University of Siena, Via Roma 56, 53100 Siena, Italy; valerio.biancalana@unisi.it; 3Department of Physics “E. Fermi”, University of Pisa, Largo Pontecorvo 3, 56127 Pisa, Italy; lorenzo.bellizzi@df.unipi.it

**Keywords:** eye tracker, vHIT, eye movements, search coil

## Abstract

Over the past 20 years, several eye-tracking technologies have been developed. This article aims to present a new type of eye tracker capable of producing detailed information on eye and head movements using an array of magnetoresistive detectors fixed on the patient’s head and a small magnet inserted into a contact lens, adapted to the curvature of the cornea of the subject. The software used for data analysis can combine or compare eye and head movements and can represent them as 2D or 3D images. Preliminary data involve an initial patient who was asked to perform several tasks to establish the accuracy, reliability, and tolerance of the magnetic eye tracker and software. The tasks included assessment of saccadic eye movements and pursuit, “drawing” alphabetic shapes or letters, and reading. Finally, a Head Impulse Test (HIT) was performed to estimate the VOR gain, comparing the standard deviation established via vHIT with that established via this magnetic eye tracker (mHIT). This prototypical device is minimally invasive, lightweight, relatively cheap, and tolerable, with a high degree of reliability and precision. All these characteristics could lead to the future use of the magnetic eye tracker in neurological and otoneurological fields.

## 1. Introduction

Eye tracking is an experimental method for recording eye movement and gaze orientation over time and in specific tasks and is a commonly used method for observing visual attention field allocation [1]. The eyes constantly move to focus and explore the environment by performing different movements that vary in range and speed, such as fixation, saccades, and pursuit, implying central and peripheral integration.

During fixation, the eyes are stable on a visual target, but since the fovea is small, the eyes must move frequently to acquire high-quality information. Fixations are relatively short (180–330 milliseconds) but can vary in length depending on several factors, such as the nature of the visual stimuli, the purpose and complexity of the task, and the ability and attention of the individual [2]. Even during a fixation, when perception appears stable, the eyes continue to move, with tremors, drifts, and microsaccades [3,4].

However, saccades can be described as ballistic movements of the eye from one fixation to the next. During saccades, visual input is suppressed, so when the eyes make a saccade, the subject is effectively blind [5,6,7]. The speed and duration of saccades are a direct function of the distance traveled [8]. Saccade measurement has become a key tool in many research fields, as the analysis of saccadic function is used to understand cognitive and visual processes [9,10] and in neurological examination and diagnosis [11,12,13].

Other important eye movements are smooth pursuit (following a moving visual target) and convergence (bringing the eyes closer or further away as a visual target is approaching or moving away from the subject). Pupil diameter and optokinetic response do not depend on voluntary control. The first is modulated by the antagonism of the parasympathetic and sympathetic nervous systems, the second is a combination of slow- phase and fast-phase eye movements: the subject chases a moving object (pursuit movement), which then leaves the visual field, and the eyes return to the original position (saccadic movement) [14]. Reorientation of the eye can also be induced in response to head movement. In otoneurological practice, the vestibulo-ocular reflex (VOR) is of great importance, which allows the eye to maintain a stable retinal image during head movements because of vestibular activations. When its functionality is reduced, it can cause dizziness and/or oscillopsia [15].

Eye-tracking technology has applications in a wide range of industries, including measuring advertising effectiveness, instrumentation to improve reading, automotive safety, driver training, accessibility interfaces, and providing objective indicators of the cognitive, psychiatric, and neurological states of individuals [16]. Abnormal eye movements can in fact be found in patients suffering from neurodegenerative diseases and their interpretation can be useful for early diagnosis. Ba et al. have shown that patients with Parkinson’s disease have worse stereopsis, with a greater number of fixations and a greater amplitude of saccades, suggesting difficulties in fixation stabilization [17]. A meta-analysis performed by Opwonya et al. revealed that prosaccade and antisaccade latencies and antisaccade error frequency show significant alterations for both mild cognitive impairment and Alzheimer’s dementia [18].

Over the past 20 years, research has developed several methods for tracking eye movements. Diverse eye-tracking devices have been proposed based on different technologies, for example, electro-oculography, infrared camera, video-oculography, and search coil: to date, the search coil technique is the gold standard in terms of speed and accuracy, however, its performance is achieved at the expense of a high degree of invasiveness [19].

This article aims to present preliminary results obtained with a new type of device capable of producing detailed information on both eye and head movements using an array of magnetoresistive detectors fixed on the patient’s head. This prototype device provides high-frequency measurements of the magnetic field produced by a small magnet inserted into a contact lens. The potential of the proposed technique is discussed here, as well as the advantages and disadvantages, in particular regarding the precision of the eye movement reconstruction, the invasiveness and tolerability for the patient, the cost, and the completeness of the information that can be extracted.

Similarly to the search coil technique, our device uses magnetic signals to determine the orientation of the eye, but the operating principle is completely different in the two cases. In fact, the search coil technique is based on the application of a time-dependent field produced by bulky external sources and its detection by a (passive) coil inserted in a contact lens; the signal is generated due to the phenomenon of Faraday induction. Our device, however, is based on a small magnet inserted into a contact lens which (actively) produces a magnetostatic field, which is then measured by sensors placed near the eye. In this case, the signal detection is based on the magnetoresistive effect.

## 2. Materials and Methods

The operating principle of the tracker is based on measuring the magnetic field in different positions near the eye. This field is made up of the superposition of an environmental term (the Earth’s field) and an artificial term, generated by a small magnet inserted into a contact lens whose curvature is adapted to the subject’s cornea.

The magnetic source is a coated disk (0.5 mm thick, 2 mm in diameter) in Neodymium-Iron-Boron (Nd-Fe-B) alloy (latest generation material for the maximum level of magnetization) and is implanted in a rigid scleral lens, a few millimeters away from the optical axis, so as not to obstruct the line of sight. It is axially magnetized and in this first implementation, it is inserted tangentially to the lens, meaning that the magnetic dipole is almost (but not perfectly) aligned with the direction of gaze (Figure 1). Minor consequences of such a misalignment are analyzed and discussed in [20], together with other interesting possible potentials (such as measuring torsional motion) that could be achieved with different arrangements, a subject beyond the scope of this paper.

The Earth’s magnetic field is homogeneous and has a typical intensity of around 30 microtesla, while the artificial field generated by the magnet is intrinsically non-homogeneous and is measured with different intensities and orientations by sensors variously positioned with respect to the magnet. The size of the magnet and the typical distance from the sensors are chosen in such a way as to also make this field inhomogeneous on the order of tens of microtesla. A superposition with similar intensities is a condition that facilitates an accurate determination of both the terrestrial field (whose components in the sensor reference system vary with the orientation of the sensor array and therefore of the patient’s head) and the inhomogeneous one (which varies when the orientation of the eyes changes relative to the sensor frame).

The sensors are placed on a rigid frame (sensor frame (SF)) so that their relative position is known. The SF dimension is of the order of a few centimeters and is shaped so as to negligibly impede vision (size and shape are similar to common goggle glasses). The SF contains eight vector sensors: each sensor measures three components of the magnetic field: each measurement provides 3 × 8 = 24 values that correspond to the three components of the magnetic field in 8 different positions around the eye (Figure 2).

For each measurement, a numerical method (best-fit procedure) is applied to convert these 24 data points into useful information. The latter is composed of 9 data that correspond to three three-dimensional vectors, namely, the position of the magnet with respect to SF, the orientation of the magnet with respect to SF and the orientation of SF with respect to the Earth’s field. The numerical procedure is performed on a regular personal computer (PC) and is fast enough to provide real-time output, even if the acquisition is performed at the maximum speed (100 samples/s, for the current implementation. About this specification, it is worth mentioning that higher rates could be achieved using similar sensors currently being developed and made available in a rapidly evolving market).

The first two vectors can be used to infer the orientation of the eye with respect to the SF (and to the patient’s head, which is rigidly connected to the SF), and the third vector allows to infer the orientation of the SF (i.e., of the patient’s head patient) with respect to the terrestrial field (this last use is completely similar to what is commonly done with compasses). In summary, numerical analysis enables the simultaneous evaluation of gaze and head orientation, since it allows both the field of the Earth and of the magnet to be separately evaluated and advantageously used.

The magnetic sensors used in this instrumentation are highly integrated and advanced devices. They are based on magnetoresistive devices, and outperform Hall sensors for measurements in the order of tens of microtesla [21]. However, these are low-cost devices owing to the fortunate circumstance of their mass production, induced by their wide application in popular electronics (e.g., to include compasses in smartphones, drones, etc.). They output digitized data at a high rate (over 100 samples/s). A second fortunate circumstance is that the speed of currently available personal computers is sufficient to perform the required numerical analysis at the same speed as to allow real-time data output [22,23,24,25].

Figure 3 outlines the operating principle of this new device: a magnet produces a non-homogeneous field, whose orientation varies in space and whose intensity decreases at increasing distances from the magnet. Different intensities and different orientations are acquired by a set of sensors. Each sensor measures specific field components depending on its position relative to the magnet. Numerical methods allow the inverse problem to be solved: from the field components measured at various known points, the position and orientation of the magnet can be deduced. If a further homogeneous field is superimposed on that magnetic field, this can also be evaluated. The latter feature provides information on the orientation of the sensor with respect to the environment and allows to track the rotations of the subject’s head. The real implementation uses eight sensors and the problem is inherently three-dimensional. Figure 3 is a simplified image only to illustrate the working principle of the tracker.

As mentioned, the software developed for data analysis provides, through a best-fit procedure, information both on the orientation of the eye with respect to the SF and on the orientation of the latter with respect to the environmental field, so that both the movements of the eyes and those of the head are evaluated simultaneously. The same software then allows a comparison of these two pieces of information to investigate the correlated eye–head movements or to combine them to deduce the absolute orientation of the gaze, of which graphic representations are then provided as 2D or 3D images.

For the preliminary in vivo tests presented in this work, the patient was asked to perform various tasks concerning the measurement of saccades, with the evaluation of the vestibular-ocular reflex (VOR) on the horizontal semicircular canals and the evaluation of pursuit movements, following a moving object displayed on a PC monitor or trace a pre-established shape, a path, or an alphabetical letter with your gaze. Another required task was to read a text on the PC monitor in order to test and evaluate the combination of all different types of movements.

Ethical approval was not required as the initial patient (a healthy 31-year-old man without vestibular disorders) is one of the authors (M.C.) and was fully informed about the aims and possible adverse effects of this research.

## 3. Results

The precision and accuracy assessment can be performed under several conditions, in both in vitro and in in vivo operations. The in vitro operation permits an evaluation of the intrinsic instrumental performance, while the in vivo experiments are inherently affected also by other practical limitations, in particular the microsaccades and tremors that cause a gaze spread around its ideal direction and have the same effect as a reduced precision. Both in vitro and in vivo experiments may be focused on angular and on angular-speed uncertainty, and both can be performed under static or dynamic conditions. Our system measure angles, both for head and for eye motion, thus when the angular speed is examined, some data filtering is necessary before the numerical estimation of the time derivative of the signal. The data filtering can be more or less strong and this introduces a degree of arbitrariness in the final estimates: an issue occurring also with other kinds of instrumentation.

Concerning the angular motion of the head, our system measures directly the angles, differing from other systems which use gyroscopic detectors that respond to angular speed and hence require time integration to retrieve angular data. The latter operation may result in a severe loss in accuracy because even a small inaccuracy in the angular speed causes a large angular error over long time intervals; such an issue is avoided in our case. The measured angles can be compared to the value measured at a given instant and relative to the SF coordinates, thus the accuracy can be assessed with respect to an essentially arbitrary reference frame, which makes the angular precision a favorite indicator of the performance. A deeper analysis of angular accuracy would require further details that have been presented in [20]. Briefly, the system cannot detect head rotation around the direction of the ambient field and eye rotations around the axis of the magnet, thus the ultimate angular accuracy depends on the specific rotations to be analyzed. We have assessed the instrumental precision both in static and dynamic experiments in the in vitro arrangements.

The instrumental angular precision of the system was assessed through in vitro static measurements that estimated the standard deviation of the magnet and of the ambient field angles in statistical samples acquired over time, while both the magnet and the sensors were maintained in fixed positions. Under these conditions, fluctuations of 0.05 degrees in sensor (i.e., head) orientation and 0.20 degrees in magnet (i.e., eye) orientation were observed, mainly originating from electronic noise and environmental magnetic disturbances.

Another estimate of the instrumental angular precision was performed dynamically. For this purpose, the magnet was positioned close to the sensors (simulating the real application case) and the sensor array was rigidly rotated around a fixed axis 1 cm away from the magnet. In this case, the reconstructed trajectory of the magnet with respect to the SF was predicted to be an arc of a circle (Figure 4A), and the standard deviation from the ideal provided an estimate of the angular uncertainty. Repeated measurements of this type resulted in an estimated uncertainty of 0.2 degrees. It is important to highlight that the mechanical simulator was designed to realistically reproduce the experimental conditions of in vivo operation.

Similarly, in vivo angular precision was derived from the analysis of tracking data. In this case, the volunteer was asked to follow with his gaze a small target (red dot) that moved slowly along a straight line on the screen of a PC provided by dedicated software, keeping his head still. The reconstructed trajectory was then fitted to a circumferential arc and the residual standard deviation of the angle from the data points to the curve was evaluated, resulting in 0.6 degrees (Figure 4B). As expected, the in vivo fluctuations exceeded the instrumental ones measured in vitro, and this increased uncertainty is to be ascribed to residual head motions (and consequent voluntary eye compensations) as well as to involuntary eye motions around the ideal direction, possibly due to tremors and microsaccades.

Several in vitro and in vivo tests were also performed to assess the accuracy in the determination of finite angular displacements. For the in vivo case, the subject was requested to look at two points separated by a known distance on a screen positioned at an assigned distance from the eyes (those points could be the vertices of a PC monitor or small figures in the monitor itself); for the in vitro case, we used the mechanical simulators described in [25]. Typical inaccuracies within 2 degrees were observed in vivo over 30- degree displacements, while in vitro, we obtained accuracies better than 1 degree over a 16-degree displacement and better than 2 degrees over a 90-degree displacement.

Another kind of assessment was aimed to characterize the dynamical response of the instrumentation to sudden (saccadic) or slow (pursuit) motion of the eye. To this end, the volunteer was asked to follow with his gaze a small target (a red dot) moving slowly or jumping back and forward on a PC screen. Figure 5 shows the estimates of angles (upper plots) and angular speeds (lower plots) in the cases of pursuits (left plots) and saccades (right plots).

The following qualitative tests represent a visual demonstration of the tracker’s accuracy, precision, and reliability. The starter patient was asked to perform different tasks that were analyzed with a dedicated software. Possible applications of the tracker might be “drawing” shapes or alphabetical letters that could eventually be used by the subject to communicate with the outer world. Hence, the first task asked of our subject was to draw a rectangle with the following results (Figure 6). 

Afterwards, the starter patient was asked to follow a red dot moving on a monitor, unaware of the movements that the dot was going to execute. The outcome of his pursuit movement was a ∞, the infinite symbol (Figure 7). 

The accuracy of the eye tracker was then evaluated for reading text. The different length of the lines is suggested in Figure 8. 

Furthermore, the subject was asked to “write” a simple word, such as “YES” in Italian (SI). The tracker was thorough and elaborated the trajectory shown in Figure 9. 

Finally, an HIT (Head Impulse Test) on the horizontal axis has been performed to establish the tracker’s reliability and accuracy in VOR (vestibular-ocular reflex) gain estimation. This reflex acts as a compensatory eye movement to keep the image stable on the retina when a head rotation is imparted: the eyes will ideally move in the opposite direction of the head, at the same speed, and line up with the axis of rotation of the head [26]. Even if the search coil is considered the gold standard for VOR measurement, this reflex is usually evaluated by vHIT (video Head Impulse Test—for this paper, by ICS Impulse—Natus), which has been demonstrated to have the same efficacy [27]. Thus the subject underwent both vHIT and “mHIT” (HIT with our magnetic eye tracker) to compare the VOR estimations provided by the two devices (Figure 10). 

For what concerns mHIT, the data analysis software provides information about both eye and head angular speed for each trial and it evaluates the VOR gain as the ratio between eye and head peak angular speeds. Alternatively, the VOR can be estimated in terms of the ratio between eye and head angular displacements [28]. The starter patient’s VOR gain measurements obtained from vHIT and mHIT included 20 trials each. The mHIT software evaluated the mean VOR gain with either the traditional slope gain (peak velocity) or the angular displacement gain. Results are summarized in Table 1 together with the standard deviation.

## 4. Discussion

Eye trackers can be used simply to record where the subject is looking and for how long he or she looked before skipping to another visual stimulus. Moreover, eye trackers can be used for interactive purposes, such as moving a cursor by quadriplegic patients [3]. Additionally, eye trackers can be implemented in VR systems to reduce the dizziness feeling also known as cybersickness produced by extensive use of these systems [29].

Eye-tracker performance is usually described in terms of accuracy and precision. Accuracy depends on how much the eye position measured by the tracker corresponds to the actual eye position, while precision is due to consistent measurements of the eye position [30]. It is important to highlight that the fovea is not strictly lined up with the magnet embedded in the contact lens, so when the software elaborates the eye movements starting from the magnet’s movements, there might be the necessity to correct them with mathematical algorithms, such as Listing’s law [31]. The details of these corrections are beyond the aim of this paper and they have been discussed in another publication [20].

This magnetic eye tracker differs from the other eye-tracking technologies for several characteristics: First of all, it does not suffer from the noise produced by facial muscle contraction and the eyes blinking, in contrast to Electro-Oculography (EOG) [32], which is an old method based on several electrodes placed around the eye to record the electric potential between the cornea and retina. As for EOG, both the devices can record eye movements, even when they are closed or the patient is non-cooperative.

The proposed device needs only a preliminary self-calibration of the sensors, after which it produces absolute angle estimations for the ambient field and for the magnet (some calibration can be necessary to convert magnet orientation to eye-gaze orientation); moreover, its functioning does not suffer from light changes or blinking, differently from IROG (Infrared OculoGraphy), which requires a patient-based recalibration and is very sensitive to external light changes, so that environment light changes can produce some biases during the data acquisition procedure [33]. IROG consists of an IR light source that illuminates the eye and an array of photodetectors that collect the reflected light [34,35], thus it cannot produce any signal when the eyes are closed and suffers from artefacts driven by eye-blinking. On the other hand, the magnetic tracker is more invasive compared to other optical devices (IROG, IR cameras) but it is not influenced by blinking.

The current gold standard for eye tracking is the scleral search coil (SCC) [36,37], a system that measures the voltages induced on a receiving coil by two or three mutually orthogonal alternating magnetic fields. These alternating fields are generated by pairs of large-size coils surrounding the subject’s head. The receiving coils are molded in a soft contact annulus that is attached by suction to the eyeball. With this method, the eye orientation is inferred based on how the scleral coil interacts with the alternating magnetic field to generate an electric signal. The bulky coils that generate the alternating field are fixed to the head or to an external frame; in any case, the head must be at their center and this constitutes a severe limitation to the patient’s movement, making the system substantially unwearable.

Despite being the gold standard for eye tracking, the SSC has the disadvantage of being highly invasive, with the risk of keratitis and corneal damage. The magnetic eye tracker here described is more comfortable than SSC, because the eye–sensor connection is inherently wireless and does not need an electric connection (the exit wire passing on the eyelid of the SSC). Additionally, the magnetic eye tracker’s sensor frame is lighter and smaller than the coil systems used to generate the alternating fields necessary for the search coil. All these features make it comfortably wearable, with negligible hindering of the head mobility and no constraints associated with the need of external field generators. Table 2 summarizes the characteristics of eye-tracker technologies presented. 

One of the SCC main expressions deals with otoneurology since it is considered the gold standard for VOR gain estimation, although vHIT has been demonstrated as comparable with it [27]. The accuracy and precision of this magnetic eye tracker are comparable to the vHIT infrared method, as inferable from the low standard deviation observed at mHIT, suggesting a possible comparison with SCC in terms of accuracy in the VOR gain estimation, with lower invasivity and the advantage of being self-calibrated. 

When performing the HIT on a wooden puppet (sensor frame rotating around a fixed vertical axis), typical results indicate an uncertainty below 5%, meaning that the ratio between head and eye angular displacements (and between angular velocities) resulted in the range of 0.95–1.05. It is worth highlighting that this instrumental error is expected to be an underestimate of the real, in vivo VOR uncertainty, where other effects (e.g., microsaccades, rotation axis instability, a non-rigid connection between head and sensors etc.) may occur, causing additional imperfections not considered in the assessment described here above [20].

Another advantage of this technology is that it can determine the position and the orientation of the eyes. This represents an innovative feature, since this device might measure torsional eye movements, a feature that the other video-based methods usually cannot do or, if they can, is often of low quality. It is important to consider that rotations around the dipole direction cannot be detected. In the current implementation, the magnet is axially magnetized, thus the magnetization is nearly parallel to the gaze directions and consequently, the system is blind to torsions, while it efficiently detects yaw and pitch rotations. Using a magnet with diametral magnetization would make the dipole transversely oriented: in such a way, the system would efficiently detect the torsions (e.g., if the dipole is transversely oriented, it would measure torsions and pitch, while it would not detect yaw). Small magnets with diametral magnetization are not commercially available; they can be produced on demand but so far we have not provided them ourselves, thus we could not test in vivo torsion detection (this will be the subject of further and already planned studies). A further possibility is using a lens with two magnets embedded. Theoretically, this might lead to a complete (three angular and three position parameters) characterization of the eye pose.

In terms of eye intrusiveness, the starter patient showed no discomfort in wearing the contact eye lens and did not require the use of any ocular anesthetic. The magnet does not obstruct the view while the sensor frame is partially obstructive. In the current implementation, the sensors are distributed on two planes and that makes the frame partially obstructive. However, we have evidence that such a two-planar distribution could be avoided and we plan to design new prototypes with reduced intrusivity.

The production cost is expected to be quite low, even if the main disadvantage of the device is that the lens must be tailored for a single subject. For this reason, other construction techniques are going to be explored, with the magnet embedded inside the lens using a 3D mold so that common soft contact lenses can be used for our purpose, avoiding the tailoring process [38].

Since the eye orientation is estimated with respect to the sensor frame while the orientation of the latter is estimated with respect to the ambient field, if the frame is not rigidly connected to the head, this feature may produce artifacts. Further developments are focused on making the sensor frame as light as possible and to improve its mechanical connection to the head.

Concerning the biocompatibility of the used materials, some care is needed to protect the magnet from chemical degradation due to long-term interaction with the lens preserving liquid. Long-lasting experiments are still ongoing to evaluate the magnet degradation and liquid contamination over time. Several kinds of organic or metallic coatings are being considered to prevent or slow down these processes, and chemical analyses are planned to provide quantitative assessments of diverse potential contaminants.

## 5. Conclusions

The proposed prototypic device aims at developing new health-oriented technology for a broad range of human well-being applications, ranging from medical diagnosis in neurological and neurotological domains to inclusive systems for severely impaired patients. The successful development of a wearable, low-invasive, lightweight, accurate, and high-frequency three-dimensional eye-tracking system will represent a breakthrough health technology enabling early detection of neurological conditions, such as multiple sclerosis, Parkinson’s or Alzheimer’s diseases, and even rare medical conditions such as progressive supranuclear palsy including cerebellar. 

The head-tracking ability of the proposed device will enable its application for diagnosis in neurotology by detecting and/or follow-up vestibular impairments, such as BBPV, vestibular migraine, or Meniere’s disease episodes. The ability to accurately and reliably estimate gaze in space will allow using the device for a range of inclusive purposes, such as developing interfaces for patients with locked-in syndromes or severe impairment in movement, who might use the technology to communicate through video terminals driven through the eye tracking, by writing some words or by linking specific eye movements to pre-established actions, as well as the ability to drive a wheelchair or other kinds of prostheses that might be moved by eye-tracking technology. Furthermore, with a feasible increase of the sampling rate, the tracker might be useful for the detection of reading disorders [39] or, more generally, to investigate subtle details in eye dynamics during reading tasks [40].

The ease of use and versatility of the device would make it suitable for integration into the health care system both by developing a standard battery of screening tests for neurological and neurotological conditions, such as stroke or midbrain pathologies to be used in the Emergency Department, and as a clinical diagnostic tool in routine procedures for neurology and neurotology visits.

This tracker is certainly more invasive than infrared oculographic systems while competing with them in terms of precision, speed, and cost-effectiveness, with the advantage of not requiring patient-based calibration. The invasivity aspect is related to the need of wearing a contact lens hosting a small magnet. 

Compared to the gold-standard technology for eye-tracking technology (based on scleral search coils), such invasive detail is dramatically reduced, due to the wireless nature of the passive magnetic target. Compared to the search-coil approach, the proposed measurement system has the advantage of being comfortably wearable, facilitating diagnostics that require head movements: the goggles hosting the sensors are small-size and lightweight and no external coils for field generators are needed. An additional feature of the proposed system is the simultaneous evaluation of both eye and head motion. This can be used to infer the absolute direction of the gaze or (with important implications in otoneurological applications) to characterize the eye response to head-motion stimuli. Besides using a magnet with diametral magnetization, the system could also efficiently detect the torsional eye movements with relatively low invasivity.

## 6. Patents

V.B. and M.M. appear as inventors in a patent pending No. WO2022018691.

## Figures and Tables

**Figure 1 brainsci-13-01439-f001:**
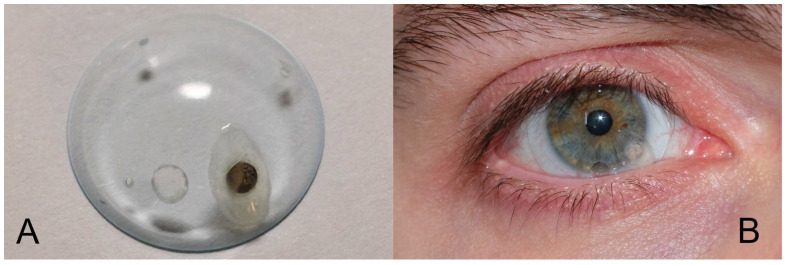
(**A**) Nd-Fe-B magnet embedded in the contact lens. (**B**) The contact lens worn by the subject.

**Figure 2 brainsci-13-01439-f002:**
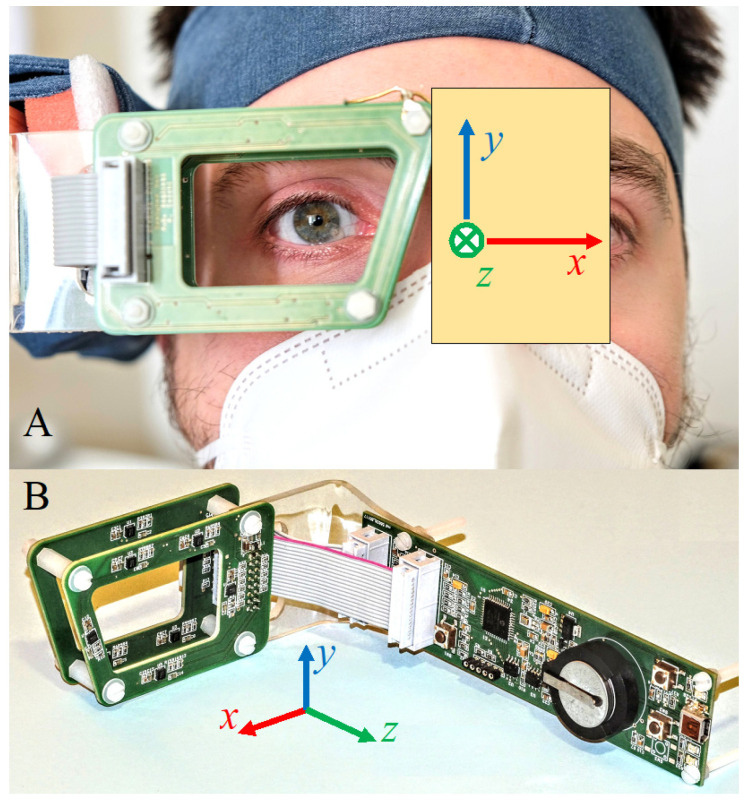
Worn sensor frame and sensors’ position. (**A**) The sensor array is rigidly fixed to the patient’s head and tracks the movements of his right eye. (**B**) It contains eight three-axial magnetoresistive sensors in two sets distributed on two parallel printed circuit boards (PCBs). Red and blue circles highlight the two sets. A third PCB hosts a microcontroller and other electronics that interface the sensors to a personal computer. The Cartesian axes with respect to the sensor array are shown in both the pictures. Nominally, when the head is erect and front-oriented, $\hat{z}$ is back-directed, $\hat{x}$ is vertical, and $\hat{y}$ is transverse-horizontal.

**Figure 3 brainsci-13-01439-f003:**
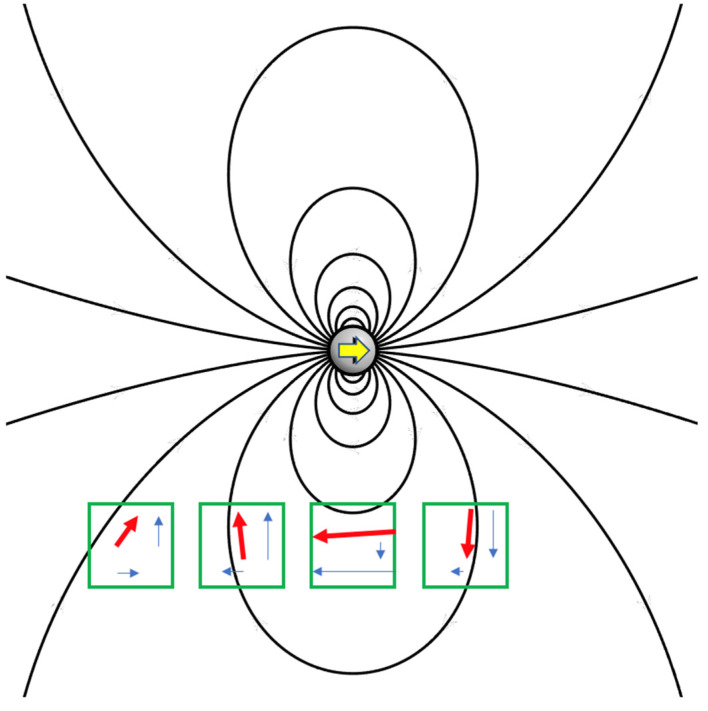
The inhomogeneous magnetic field produced by a magnet (yellow arrow). The orientation varies in space and strength depending on the distance from the magnet. The black lines describe the field orientation and the red arrows represent a sample of field measurements performed in a set of locations where a set of sensors (green boxes) are located. The blue arrows highlight the vectorial nature of the measured quantities. The schematics in this figure simplify the real condition, where the sensors are 8 (not 4) and each of them measures 3 (not 2) components of the field. Furthermore, an additional homogeneous field (the Earth’s field) sums to those illustrated.

**Figure 4 brainsci-13-01439-f004:**
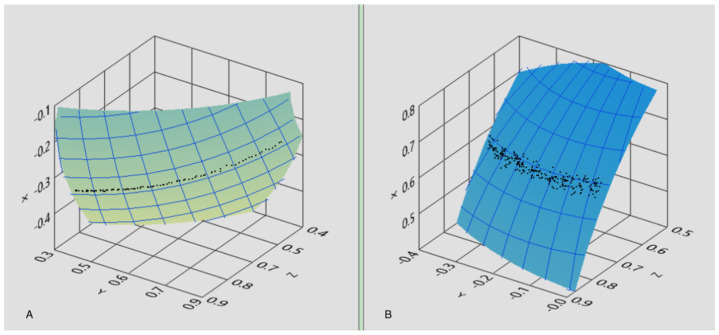
Estimated uncertainty of the system in dynamic conditions: the reconstructed trajectory of the magnet orientation is expected to be an arc of circumference and the standard deviation from ideality provides an estimate of angular uncertainty. (**A**) Estimated uncertainty was measured by rotating the sensor’s frame around a fixed axis with a mechanic simulator. (**B**) Estimated uncertainty was measured on the starter patient wearing the lens.

**Figure 5 brainsci-13-01439-f005:**
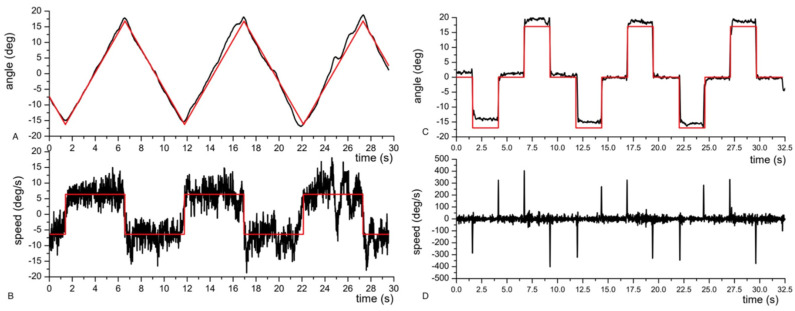
Analysis of pursuit (**A**,**B**) and saccades (**C**,**D**). Red line = ideal trajectory followed by a small target moving on the PC screen. Black line = trajectory reconstructed by the magnetic eye tracker. (**A**) Comparison between the eye and the target’s angular displacement during pursuit. (**B**) Comparison between the eye and the target’s angular speed. (**C**) Comparison between the eye and the target’s angular displacement during saccades. (**D**) Eye’s angular variation speed during saccades.

**Figure 6 brainsci-13-01439-f006:**
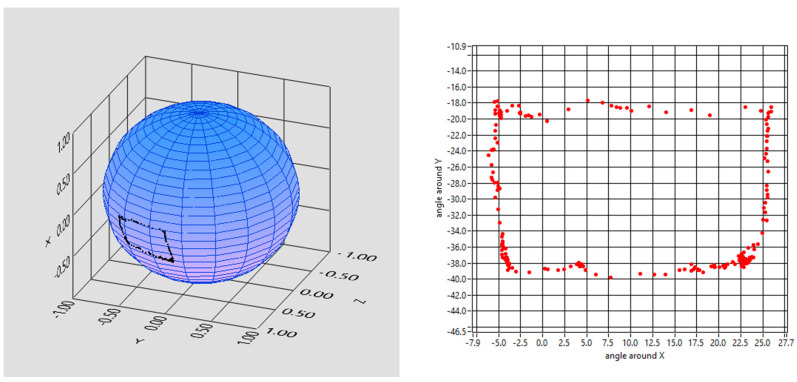
Outcome of drawing a rectangle and reconstruction in 3D and 2D.

**Figure 7 brainsci-13-01439-f007:**
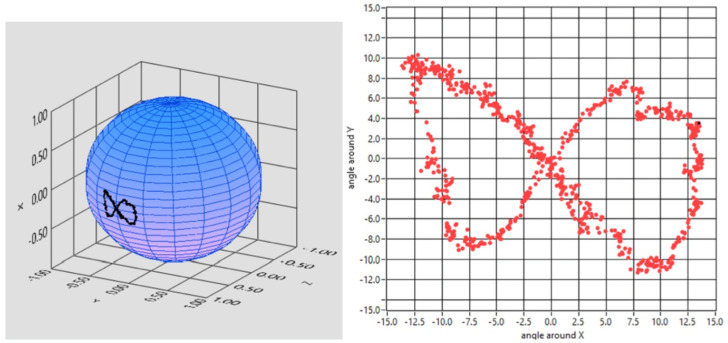
Reconstruction of the eye tracking while the patient was following a moving dot on the monitor. The total pathway was the infinite symbol, here edited in 3D and 2D.

**Figure 8 brainsci-13-01439-f008:**
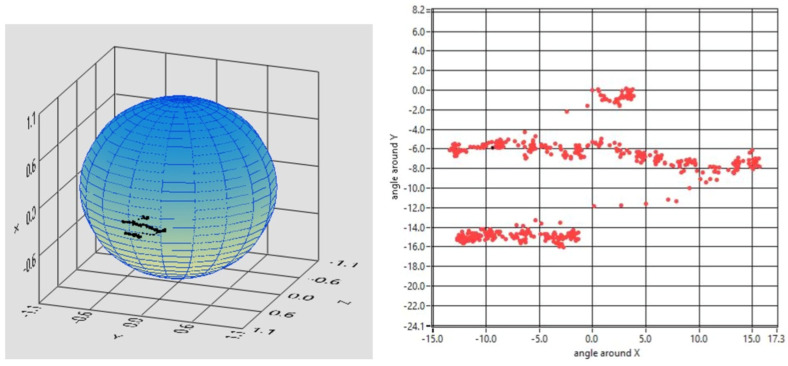
The starter patient was asked to read aloud three lines of text. The tracker recognizes the different position and length of the lines. The dynamics can be studied as well. In this case, the subject spent 1920 ms, 8100 ms, and 4560 ms to read the three lines in sequence, which contain 22, 90, and 36 letters, respectively (the reading rates are 11.5, 11.1, and 8 chrs/s, respectively). The saccades leading to the beginning of the next lines are resolved as well (here, one point at each 20 ms is visualized).

**Figure 9 brainsci-13-01439-f009:**
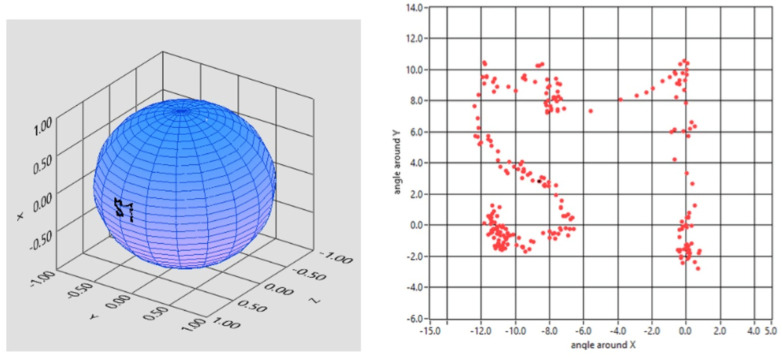
The subject writes the Italian word for YES, “SI”.

**Figure 10 brainsci-13-01439-f010:**
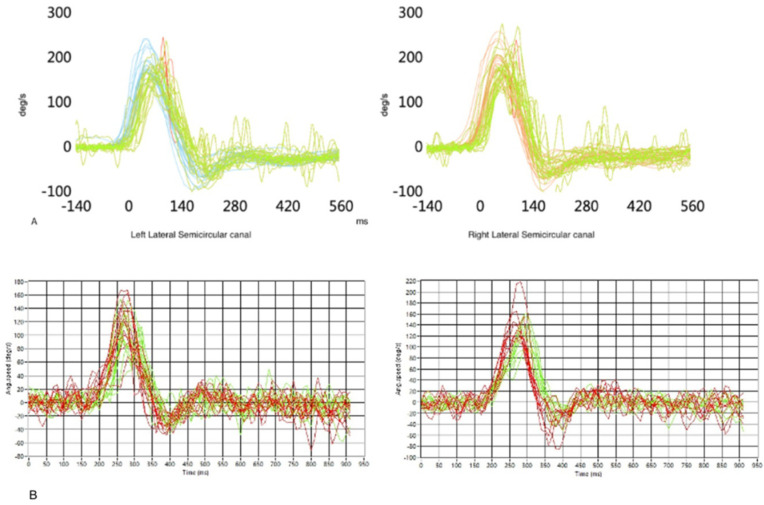
(**A**) Reconstruction and evaluation of the VOR gain of the starter patient with vHIT for lateral semicircular canals. Green line = VOR. Red line = saccades. Blue line = head movement in left vHIT. Orange = head movement in right vHIT. (**B**) Same measurements performed with the magnetic eye tracker. Green line = VOR. Red line = head movement.

**Table 1 brainsci-13-01439-t001:** VOR gain estimation with a commercial instrumentation (vHIT) and with the described device (mHIT). The mHIT provides mean VOR gain measurements slightly higher than the vHIT likely due to a non-rigid connection between head and sensors and/or some lens slippage with respect to eye surface. Nevertheless, the standard deviations (σ) in all three scenarios are similar, supporting the reliability of the magnetic eye-tracker measurements.

	Mean VOR Right LSC	σ	Mean VOR Left LSC	σ
**vHIT**	0.95	0.06	0.87	0.06
**mHIT** ^1^ Position Gain (Angles)	0.87	0.15	0.97	0.15
**mHIT** ^1^ Slope Gain (Velocity)	0.89	0.16	0.84	0.16

^1^ mHIT obtained through this prototypic device.

**Table 2 brainsci-13-01439-t002:** Main features of currently available eye-tracking technologies.

Eye Tracker	Technology	Advantages	Disadvantages
EOG (Electro-Oculography)	Several electrodes placed around the eye to record the electric potential between cornea and retina.	Can record movements of closed eyes (even when the patient is not cooperative).	Noise produced by contraction of facial muscles and eyes blinking.
IROG (Infrared OculoGraphy)	IR light source that illuminates the eye + array of photodetectors (or video camera) which collect the light reflected towards.	Not invasive.	Patient-based recalibration. Sensitive to external light changes (environment light can produce bias).
SSC (Scleral Search Coil)	2–3 pairs of large-size coils mounted in a cubic frame that produce alternating fields inducing electric signals in the scleral coil.	Current gold standard (high accuracy and precision). Mature technology, developed since 1963 [36,37].	Requires special corneal lenses. Highly invasive, with risk of keratitis and corneal damage. Needs an external field generator.
Magnetic Eye Tracker	8 sensors placed on a rigid frame: each sensor measures the magnetic field produced by a magnet embedded in a contact lens. Numerical method provides information about the magnet position.	High accuracy and precision. Wireless sensor connection. Can record closed eye movements. One-time calibration of the lens. Does not depend on environmental light.	Requires special corneal lenses. Invasive for the presence of the lens (but reduced invasivity compared to SSC). Currently available as a prototype: further development needed.

## Data Availability

Available upon request to the authors.

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
