# Peer review of "Development of a Magnetoresistive-Based Wearable Eye-Tracking System for Oculomotor Assessment in Neurological and Otoneurological Research—Preliminary In Vivo Tests"

_brainsci, 2023, doi:10.3390/brainsci13101439_

Round 1

Reviewer 1 Report

The authors introduce a new device for eye-tracking that has the potential for future research and clinical applications. I am not a physicist or material scientist, so I cannot comment on the device itself. My comments will focus on other aspects of the manuscript. Overall, I believe this eye-tracker has significant potential, but there are some concerns that need careful attention.

My primary concern pertains to the precision and accuracy of eye position measurements using this eye-tracker (please refer to this video, starting at 13:24, discussing 'Accuracy and Precision': https://www.youtube.com/watch?v=KJrnG4Tw-74). I have been using various eye-trackers for many years, primarily Eyelink video-based eye trackers, but I have also used search-coil and Dual Purkinje eye tracking systems. As described by the authors, I do recognize the potential advantages of using this particular eye tracker. However, the methodology employed to test the precision and accuracy of this eye tracker remains unclear. It would be prudent to conduct a comprehensive battery of tests to systematically evaluate its precision and accuracy. The manuscript provides vague descriptions of this information in the results section. There is a lack of quantitative measurements regarding the accuracy and precision of this eye-tracker in a battery of tests with real participants, such as step saccade and smooth pursuit tasks. While some measurements, like VOR gain, are mentioned in Figure 10 and Table 1, the process for deriving these estimations is not clearly explained.

Similarly, Figure 8 presents data from reading; however, the authors do not provide the distribution of fixation durations (e.g., first fixation durations, gaze durations) and saccade lengths (in letter unit). Including these results for reading or picture viewing (as well as other standard tasks such as step, gap tasks) could assist readers in comparing their findings with those obtained using other commonly used eye-trackers. This would enhance the validity of this eye-tracker.

I believe it is crucial to recruit a larger number of participants to validate the performance of this eye-tracker. The authors noted that 'in this first implementation it is inserted tangentially to the lens, which means the magnetic dipole is nearly (but not perfectly) aligned with the gaze direction' (line 96). This suggests that there may be a certain degree of alignment variability associated with the use of this eye-tracker. It is essential to consider how this alignment variability might impact the precision and accuracy of the eye tracker. Increasing the number of participants in the testing process would undoubtedly contribute to demonstrating the reliability of this eye-tracker.

The authors occasionally employ a more conversational language, as seen in phrases like 'thanks to the vestibular activations' and 'thanks to magnetoresistive effect.' Such informal language is less typical in academic writing.

What is the sampling rate of this eye-tracker?

The authors should exercise caution when describing this eye-tracker. As far as I am aware, there are several advantages to using this eye-tracker, and it is notably more user-friendly than the search coil eye-tracker. However, it's essential to acknowledge that this eye-tracker is still considered an invasive method for eye tracking.

Typo: line 304 “it is not affected by…”

believe the manuscript would benefit from language editing.

Author Response

Report 1

1.1 The authors introduce a new device for eye-tracking that has the potential for future research and clinical applications.
This preliminary statement correctly highlights that the manuscript describes the in-vivo application of a new device: this is the actual status of the research. As also reported in the abstract “preliminary data are presented”: no extensive clinical trials have been performed, so far.

I am not a physicist or material scientist, so I cannot comment on the device itself. My comments will focus on other aspects of the manuscript. Overall, I believe this eye-tracker has significant potential, but there are some concerns that need careful attention.

1.2 My primary concern pertains to the precision and accuracy of eye position measurements using this eye-tracker (please refer to this video, starting at 13:24, discussing 'Accuracy and Precision': https://www.youtube.com/watch?v=KJrnG4Tw-74). I have been using various eye-trackers for many years, primarily Eyelink video-based eye trackers, but I have also used search-coil and Dual Purkinje eye tracking systems. As described by the authors, I do recognize the potential advantages of using this particular eye tracker. However, the methodology employed to test the precision and accuracy of this eye tracker remains unclear. It would be prudent to conduct a comprehensive battery of tests to systematically evaluate its precision and accuracy. The manuscript provides vague descriptions of this information in the results section. There is a lack of quantitative measurements regarding the accuracy and precision of this eye-tracker in a battery of tests with real participants, such as step saccade and smooth pursuit tasks. While some measurements, like VOR gain, are mentioned in Figure 10 and Table 1, the process for deriving these estimations is not clearly explained.

We thank the reviewer for pointing out this lack of clarity and consequent weakness of the manuscript. In the revised version we have added several more explicit sentences to better explain the procedures followed to assess precision and accuracy. The matter cannot be simply summarized, because one has to deal with either accuracy or precision, either angles or angular speed data, with either head or eye motion, with either in-vitro or in-vivo measurement. The question about accuracy is pretty simple when angular velocity is considered (for which other issue occur due to ne need of additional numerical treatments) but is more complicate for angles. In fact our system measure angles, it does not need time-integration and hence is not affected by possible drifts that would deteriorate in time the accuracy. On the other hand, the eye angle is measured with respect to the sensor-frame, which has not an absolute meaning: in conclusion only the precision turns out to be a valuable indicator for the uncertainty performance.

1.3 Similarly, Figure 8 presents data from reading; however, the authors do not provide the distribution of fixation durations (e.g., first fixation durations, gaze durations) and saccade lengths (in letter unit). Including these results for reading or picture viewing (as well as other standard tasks such as step, gap tasks) could assist readers in comparing their findings with those obtained using other commonly used eye-trackers. This would enhance the validity of this eye-tracker.
We have inserted the missing information about time dynamics in the caption of the figure.

1.4 I believe it is crucial to recruit a larger number of participants to validate the performance of this eye-tracker. The authors noted that 'in this first implementation it is inserted tangentially to the lens, which means the magnetic dipole is nearly (but not perfectly) aligned with the gaze direction' (line 96). This suggests that there may be a certain degree of alignment variability associated with the use of this eye-tracker. It is essential to consider how this alignment variability might impact the precision and accuracy of the eye tracker. Increasing the number of participants in the testing process would undoubtedly contribute to demonstrating the reliability of this eye-tracker.

This remark is very important and this is confirmed by the fact that it shares objection expressed also by Rev.2 (please see also the the remark R2.1 and our corresponding reply) As said, an extensive trial over a multitude of patients is an evident a goal of our research, however it is not at the focus of our manuscript because it is beyond the current development level of the study. Indeed the actual status of our activity is at the preliminary tests in-vivo which follow other analyses mainly performed in vitro and previously published in more technical papers. The latter were aimed to describe the instrumentation and the data elaboration rather than the applicability in the field. In those papers, the performance in terms of instrumental precision and accuracy were discussed, as well.

1.5 The authors occasionally employ a more conversational language, as seen in phrases like 'thanks to the vestibular activations' and 'thanks to magnetoresistive effect.' Such informal language is less typical in academic writing.

We took care of these imperfections and we have also performed a more general revision of the English.

1.6 What is the sampling rate of this eye-tracker?

The current implementation of the tracker has a data acquisition rate firmware-limited at 100 Sa/s. The used magnetoresistive ICs would allow a maximum rate as high as 200Sa/s. It is worth noting that versions of similar sensors were available, for which the rate could be increased up to 1kSa/s (with a bottle-neck at the data transmission, which would make necessary to acquire bursts of buffered data). In summary, for some applications, similar devices could be designed to record data at much higher rate, but in bursts lasting few tens of milliseconds. The production of those faster sensors seems currently discontinued, but other producers might have similar ones in catalogue. We have added a sentence about this topic.

1.7 The authors should exercise caution when describing this eye-tracker. As far as I am aware, there are several advantages to using this eye-tracker, and it is notably more user-friendly than the search coil eye-tracker. However, it's essential to acknowledge that this eye-tracker is still considered an invasive method for eye tracking.

The invasiveness of this tracker is definitely higher than any device based on optical detection (i.e. IR-cameras and IR-reflection devices) and this actually said and summarized in the Sec.V (This tracker is certainly more invasive than infrared oculographic systems while competing with them in terms of precision, speed and cost-effectiveness...) . Compared to the search-coil apparatuses, our proposed device come with some advantages related mainly to two aspects. First, no electrical parts or electrical connections are needed in the lens. Second no AC-field-generators are needed around (or fixed to) the patient head. The first point lowers the invasiveness level, and the second one increases the wearability. Of course more research is needed to effectively compete with a mature and extensively developed technique such as the search-coil one, as well as to carefully point out and adequately compare weaknesses and potential.

1.8 Typo: line 304 “it is not affected by…”

Thanks, we have fixed it.

Reviewer 2 Report

see attached pdf

Author Response

Report 2

The scientific article titled "A Wearable Magnetic Eye Tracker" presents an innovative and promising development in the field of eye-tracking technology. This review evaluates the strong points and points for consideration in this research paper.

Strong Points:

P2.1. Innovative Technology: The article introduces a novel eye-tracking technology that utilizes a magnetic approach. Unlike traditional eye-tracking devices that rely on cameras, this wearable magnetic eye tracker employs an array of magnetoresistive detectors fixed on the patient's head and a small magnet inserted into a contact lens customized to the subject's cornea curvature. This innovation presents a fresh perspective in the field.

P2.2. Comprehensive Data Analysis: The paper highlights the capabilities of the software used for data analysis, which can effectively combine and compare both eye and head movements. Additionally, it can generate 2D or 3D images to visualize the recorded data. This comprehensive approach to data analysis is a significant asset for researchers and clinicians.

P2.3. Experimental Rigor: The study demonstrates a high level of experimental rigor by subjecting the magnetic eye tracker and software to a series of tasks. These tasks include evaluating saccadic eye movements, pursuit, shape and letter drawing, reading, and the Head Impulse Test (HIT) for VOR gain estimation. Such a thorough evaluation establishes the accuracy, reliability, and tolerance of the technology.

P2.4. Practicality: The article emphasizes the practicality of the device. The magnetic eye tracker is described as low invasive, lightweight, relatively low-cost, and tolerable by subjects. These attributes make it a viable option for various applications within neurological and otoneurological fields.

P2.5. Future Potential: The paper concludes by suggesting potential future applications of the magnetic eye tracker in neurological and otoneurological research. This device's high reliability, accuracy, and practicality indicate its potential to make valuable contributions to these fields.

We greatly appreciate the positive and encouraging assessment expressed by Rev.2 in his comments P2.1-P2.5. We thank him for his/her careful reading and for considering the potential of the proposed instrumentation, despite its not yet mature level of development. It is clear that further steps are needed both to technically refine the device and to test it extensively in field application, nevertheless we strongly believe that the preliminary results obtained so far seem already worth of being shared with the interested community and Rev.2 agree with us in this evaluation. In several instances of this new version we have stressed the current status of our work, mentioning the preliminary nature of the in-vivo tests presented, underlining an information that was already present in the abstract.

Points for Consideration:

R2.1. Sample Size: The article mentions preliminary data based on a single "starter patient." While this is a reasonable starting point for proof of concept, it is essential to conduct more extensive studies with larger and diverse subject groups to validate the technology's broader applicability.

We fully agree with this remark, which confirms a similar one expressed by Rev.1. Extending the trial to a larger set of patients (possibly including non-healthy subjects) is definitely among the future tasks of our project. However, at the current status of our research, we aim to demonstrate the feasibility rather than to provide a statistical assessment of the proposed methodology. Following more extensive sets of in-vitro characterizations, the here presented, preliminary in-vivo tests permit to confirm some positive features as well as to point out issues that did not emerge in vitro (e.g. prevent lens and sensor slippage, finding providers that may effectively produce tailored lenses, test tolerability ad chemical stability overt time etc.). The matter of performing trial on a larger set of patient is in the agenda, but for future (and possibly non-immediate) activity.

R2.2. Long-term Tolerance: The paper asserts that the magnetic eye tracker is tolerable, but additional data on the device's long-term comfort and potential side effects, especially when used over extended periods, would be valuable.

Concerning the biocompatibility of the used materials, some care is needed to protect the magnet from chemical degradation due to long-term interaction with the lens preserving liquid. Long-lasting experiments are still ongoing, to evaluate the magnet degradation and liquid contamination over time. Several kinds of organic or metallic coatings are being considered to prevent or slow down these processes, and chemical analyses are planned to provide quantitative assessments of diverse potential contaminants.

R2.3. Comparative Data: While the article mentions comparing the magnetic eye tracker to the standard deviation established through vHIT (Video Head Impulse Test), it would be beneficial to provide more detailed comparative data and statistical analysis to highlight the advantages of the new technology over existing methods.

The starter patients underwent 15 impulses for each side, both for vHIT and “m-HIT”. The mean VOR gain evaluation is represented in table 1. Details has been discussed in a previous paper: Reference 20 (A wearable wireless magnetic eye tracker, in vitro and in vivo tests), published in a trade journal.

R2.4. Title: The title is very simplistic and resembles one of popular science magazines. Please, change it to something more scientific like: "Development and Validation of a Magnetoresistive-Based Wearable Eye Tracking System for Comprehensive Oculomotor Assessment in Neurological and Otoneurological Research".

Thank you for the suggestion

R2.5. You can place the following reference in #40 in the references list which is now empty:

Thank you, the reference has been fixed

In conclusion, "A Wearable Magnetic Eye Tracker" introduces an exciting and innovative technology with the potential to revolutionize eye-tracking applications in neurological and otoneurological research. The paper demonstrates strong points such as innovation, comprehensive data analysis, experimental rigor, practicality, and future potential. However, it should consider addressing the sample size, long-term tolerance, and providing more extensive comparative data to strengthen its claims further.

Thus, I recommend minor revisions.

We thank again the Reviewer for his/her overall positive evaluation of our work and for his/her constructive remarks.

Round 2

Reviewer 1 Report

The authors have addressed most of my comments, but some of them have not been fully resolved.

Firstly, while the term "preliminary" has been used in the abstract and the introduction, the title of the manuscript is "Development and Validation of a Magnetoresistive-Based Wearable Eye Tracking System for Comprehensive Oculomotor Assessment in Neurological and Otoneurological Research." This title could be somewhat misleading, considering that only preliminary data are presented in this manuscript, making it challenging to claim full validation of the eye tracker. I understand that this manuscript primarily focuses on detailing the method of this new device, and validation might be the focus of other future manuscripts. To address this concern, I recommend that the authors explicitly acknowledge this limitation in their conclusion and emphasize the need for a more extensive validation with a larger number of participants. As they mentioned in their rebuttal, "an extensive trial over a multitude of patients is an evident goal of our research, however, it is not the primary focus of our manuscript because it is beyond the current development level of the study. Indeed, the actual status of our activity is at the preliminary tests in-vivo..."

Secondly, the authors have partially addressed my concern regarding fixation durations or saccade length, particularly in their reading results (Figure 8). They now report overall reading times per line, which is not what I was requesting. I would like to see the mean fixation durations (not just total viewing times per line), saccade lengths (in letter units), and their durations. This information is crucial, especially considering the relatively low sampling rate of the current version of this eye tracker (100 Hz). It's important to demonstrate that the implemented algorithms can reliably identify individual fixations and saccades during reading or other tasks.

n/a

Author Response

We have taken into account the important remarks expressed by Rev1: we thank him/her again for the constructive feedback and for the extremely fast response

remark 1:
Firstly, while the term "preliminary" has been used in the abstract and the introduction, the title of the manuscript is "Development and Validation of a Magnetoresistive-Based Wearable Eye Tracking System for Comprehensive Oculomotor Assessment in Neurological and Otoneurological Research." This title could be somewhat misleading, considering that only preliminary data are presented in this manuscript, making it challenging to claim full validation of the eye tracker. I understand that this manuscript primarily focuses on detailing the method of this new device, and validation might be the focus of other future manuscripts. To address this concern, I recommend that the authors explicitly acknowledge this limitation in their conclusion and emphasize the need for a more extensive validation with a larger number of participants. As they mentioned in their rebuttal, "an extensive trial over a multitude of patients is an evident goal of our research, however, it is not the primary focus of our manuscript because it is beyond the current development level of the study. Indeed, the actual status of our activity is at the preliminary tests in-vivo..."

authors’ reply:
Thanks, we have changed the title in “Development of a Magnetoresistive-Based Wearable Eye Tracking System for Oculomotor Assessment in Neurological and Otoneurological Research - Preliminary In Vivo Tests.”

remark 2:
Secondly, the authors have partially addressed my concern regarding fixation durations or saccade length, particularly in their reading results (Figure 8). They now report overall reading times per line, which is not what I was requesting. I would like to see the mean fixation durations (not just total viewing times per line), saccade lengths (in letter units), and their durations. This information is crucial, especially considering the relatively low sampling rate of the current version of this eye tracker (100 Hz). It's important to demonstrate that the implemented algorithms can reliably identify individual fixations and saccades during reading or other tasks.

authors’ reply:
The figure below is the track (angle versus time) recorded during the reading task (the central, longer line in Fig.8). Although some saccades and fixations can be observed (a few of them are indicated by red and blue arrows, respectively) such a measurement is not sufficient to quantitatively assess the saccade length (in letter units) and their duration. However it is important to stress that the measurements were focused on the system performance to estimate the gaze trajectory during a dynamic task, rather than to a saccade targeting evaluation. As the sampling rate could be doubled even with the sensors currently used, and more specific reading tasks can be designed to evaluate saccades and fixations, we are confident that these quantities (barely appreciable in this figure) can be studied with a much improved accuracy. This interesting application definitely deserves additional efforts, and we made it explicit in the conclusion section.